# AU-Harness: An Open-Source Toolkit for Holistic Evaluation of AUdio-LLMs

## Abstract

Large Audio Language Models (LALMs) are rapidly advancing, but evaluating them remains challenging due to inefficient toolkits that limit fair comparison and systematic assessment. Current frameworks suffer from three critical issues: slow processing that bottlenecks large-scale studies, inconsistent prompting that hurts reproducibility, and narrow task coverage that misses important audio reasoning capabilities. We introduce **AU-Harness**, an efficient and comprehensive evaluation framework for LALMs. Our system achieves a speedup of up to 127% over existing toolkits through optimized batch processing and parallel execution, enabling large-scale evaluations previously impractical. We provide standardized prompting protocols and flexible configurations for fair model comparison across diverse scenarios. Additionally, we introduce two new evaluation categories: LLM-Adaptive Diarization for temporal audio understanding and Spoken Language Reasoning for complex audio-based cognitive tasks. Through evaluation across 380+ tasks, we reveal significant gaps in current LALMs, particularly in temporal understanding and complex spoken language reasoning tasks. Our findings also highlight a lack of standardization in modalities of user-provided instructions existent across audio benchmarks, which can lead to performance differences of up to 7.1 absolute points on challenging complex instruction following downstream tasks. AU-Harness provides both practical evaluation tools and insights into model limitations, advancing systematic LALM development. [1]

## 1 Introduction

The emergence of Large Audio Language Models (LALMs) has opened new frontiers, extending capabilities beyond textual inputs to speech, sounds, and multimodal inputs (Tang et al., 2023; Cui et al., 2024). This progress has accelerated the development of frontier LALMs and audio-focused benchmarks. Recent multimodal LALMs like Gemini 2.5 (Comanici et al., 2025), Qwen2.5-Omni (Xu et al., 2025) have demonstrated substantial audio understanding capabilities well beyond the traditional Automatic Speech Recognition (ASR) tasks. However, despite these advances, audio evaluation toolkits have received comparatively little attention. Thus, there is a need for efficient, customizable, and consistent evaluation frameworks for fair model comparisons which can evolve as audio tasks and model complexities grow.

Existing efforts including AIR-Bench (Yang et al., 2024), AudioBench (Wang et al., 2025a), Kimi-Eval (Ding et al., 2025), and DynamicSUPERB-2.0 (Huang et al., 2024b) have broadened task coverage from ASR to spoken question answering and scene understanding. However, prevailing toolkits still face three persistent limitations. First, **throughput**: many pipelines under-utilize batching and parallelism, creating bottlenecks that preclude large-scale, systematic comparisons. Second, **reproducibility**: ad-hoc prompting and non-standardized input formatting lead to evaluation variance across setups. Third, **task scope**: evaluations rarely probe prompted temporal understanding (e.g., diarization) or spoken reasoning with unified, reproducible protocols.

Most current evaluation frameworks depend on simplistic yet inefficient input processing pipelines that struggle to scale with the increasing volume and complexity of audio benchmarks and LALMs. These limitations not only constrain the throughput of large-scale evaluations but also hinder fair and

---

[1] We will open-source AU-Harness upon acceptance to encourage future audio research.

reproducible comparisons across models of different sizes and architectures. As the field progresses toward more diverse and challenging audio tasks, the shortcomings of current evaluation infrastructure may pose a critical bottleneck, ultimately hampering the potential progress of LALMs. Unlike previous evaluation frameworks, we introduce an efficient vLLM batching orchestration together with effective data sharding to scale the evaluations across multiple nodes and hardware architectures, leading to improved efficiency for audio benchmark evaluations.

Beyond computational efficiency, existing toolkits also suffer from a notable lack of customizable configurations for different audio task configurations, severely limiting their utility for diverse research and application needs. Insufficient attention to task-specific prompting remains a significant challenge for LALM evaluation and comparison across different benchmarks. Prompt sensitivity further compounds customizability concerns, since LALMs' outcomes might significantly change when slight variations in prompt phrasing (Cui et al., 2024).

In addition, audio benchmarks and evaluation kits remain restricted to spoken language understanding and observational audio reasoning (analyzing audio content). To advance toward practical applications, we focus on *operational spoken reasoning* [2] tasks that require executing instructions delivered through speech, such as function calling, code generation, and complex instruction following. This complements existing benchmarks like MMAR (Ma et al., 2025), MMAU-PRO (Kumar et al., 2025), and MMSU (Wang et al., 2025b) that evaluate observational reasoning over mixed audio scenes. We construct and integrate *operational spoken reasoning tasks* with our evaluation kit for comprehensive agentic audio-to-text generation support. In addition, we also provide support for LLM-adaptive diarization evaluation where LLM prompting results in different I/O formats. To the best of our knowledge, our proposed evaluation kit is the first to introduce operational spoken reasoning tasks and support LLM-Adaptive Diarization evaluations.

Our contributions are as follows:

- We propose an **efficient evaluation engine** that leverages vLLM batching and dataset sharding to scale evaluations to multi-node infrastructures without sacrificing fidelity.

- A **unified, configurable framework** that standardizes prompting and metrics across benchmarks, enabling fair, reproducible comparisons and easy task integration.

- **Expanded evaluation coverage** with LLM-Adaptive Diarization and Spoken Language Reasoning to assess temporal grounding and audio-conditioned reasoning in LALMs.

## 2 RELATED WORK

**Audio Benchmarks.** Benchmarks play a critical role in the development of LALMs. SUPERB (Yang et al., 2021) established core task axes (Content, Speaker, Semantics, Paralinguistics) for audio model evaluation. DynamicSUPERB (Huang et al., 2024b) and DynamicSUPERB-2.0 (Huang et al., 2024a) expanded coverage to instruction-tuned and sequence generation tasks across speech, music, and environmental audio. Instruction-following and agentic behaviors have been probed by AIR-Bench (Yang et al., 2024) and VoiceBench (Chen et al., 2024). More recently, AudioBench (Wang et al., 2025a) unified eight task families over 26 datasets for AudioLLMs.

Complementary efforts in 2025 broaden the breadth and depth with *observational audio reasoning*: X-ARES (Zhang et al., 2025) systematically assesses general audio encoders across domains, AHELM (Lee et al., 2025) aggregates multi-aspect evaluation for audio-language models (reasoning, robustness, safety, multilinguality), MECAT (Niu et al., 2025) targets fine-grained audio understanding with expert-guided captions and QA. MMAR (Ma et al., 2025), MMAU-PRO (Kumar et al., 2025), and MMSU (Wang et al., 2025b) focus on understanding and analyzing complex audio scenes, spatial relationships, and mixed-audio reasoning. CodecBench (Wang et al., 2025c) benchmarks codecs from acoustic and semantic perspectives. While these benchmarks excel at observational tasks, few evaluate *operational spoken reasoning* where models must execute tasks through speech instructions, or prompted diarization with reproducible protocols. This gap motivates our focus on operational reasoning capabilities like function calling, code generation, and complex in-

---

[2]Throughout this work, we use the term *Spoken Language Reasoning* to refer to our pre-defined *Operational Spoken Reasoning*, unless stated otherwise.

struction following delivered through speech, complementing the observational reasoning emphasis of existing benchmarks.

**Audio Evaluation Kits.** In contrast with Audio Benchmark development, Audio Evaluation Kits have received less attention. This can be primarily attributed to the straightforward nature and minimal setup requirements of the early audio tasks, as presented in Huang et al. (2024b) and Yang et al. (2024). However, the rapid growth of LALMs and the increasing complexity of newly curated audio benchmarks have underscored the critical need for comprehensive evaluation kits, as exemplified through the development of extensive evaluation kits (Ding et al., 2025; Wang et al., 2025a; Chen et al., 2024). For instance, AudioBench (Wang et al., 2025a) offers versatile evaluation support for up to 8 tasks across 26 datasets. VERSA (Shi et al., 2025) introduces a comprehensive framework to evaluate the quality of various speech, audio and music signals, with the focus on text-to-audio applications. Despite these advancements, most current evaluation kits operate on the simplified assumption that *a single model is evaluated against a single benchmark per run*. Addressing this limitation, we introduce an efficient, customizable evaluation kit to support the massive growth of the current LALMs and audio benchmarks as summarized in Table 1.

## 3 LALM EVALUATION CHALLENGES

Table 1: **Feature comparison of contemporary LALM evaluation toolkits.** We evaluate key technical capabilities across existing frameworks: multilingual support for evaluating models across diverse languages, vLLM integration for efficient batching, multi-turn dialogue support for conversational scenarios, LLM-Adaptive Diarization for temporal understanding through prompting, Spoken Language Reasoning for complex audio-conditioned cognitive tasks, and configurable prompt customizations for flexible evaluation design. Our framework is the first to provide comprehensive support across all dimensions.

| EvalKit | Multilingual Support | vLLM support | Multi-turn | LLM-Adaptive Diarization | Spoken Language Reasoning | Configurable Prompt Customizations |
|---|---|---|---|---|---|---|
| AudioBench | ✓ | ✗ | ✗ | ✗ | ✗ | ✗ |
| Kimi-Eval | ✓ | ✗ | ✗ | ✗ | ✗ | ✗ |
| VoiceBench | ✓ | ✗ | ✗ | ✗ | ✗ | ✗ |
| AU-Harness | ✓ | ✓ | ✓ | ✓ | ✓ | ✓ |

### 3.1 INFERENCE EFFICIENCY

Existing LALM evaluation kits have been designed based on the assumption that *a single model should be evaluated against a single benchmark per run*. However, this constrains researchers from conducting systematic, large-scale comparisons across LALMs and audio benchmarks efficiently, slowing the iterative process of model development and refinement. The current evaluation kits also under-utilize parallel processing capabilities available in the high-performance computing clusters, resulting in failures in incorporating benefits of available hardware infrastructures.

Two essential task-agnostic metrics for evaluating the efficiency of LALM evaluation frameworks are Real-time Factor (RTF) and Processed Samples per Second. RTF measures the processing time of an evaluation framework rel-

Table 2: **Throughput efficiency comparison across LALM evaluation frameworks.** Results averaged over 500 samples from LibriSpeech-test-clean (1.05 hours total audio). Real-time Factor (RTF, ↓ better) measures processing time relative to audio duration. Processed Samples per Second (↑ better) quantifies raw throughput. Our framework achieves 48.75% RTF reduction and 95.19% throughput increase over the best competing baseline, demonstrating substantial efficiency gains through vLLM integration and request orchestration.

| EvalKit | RTF (↓) | Processed Samples per Second (↑) |
|---|---|---|
| AudioBench | 19.9 | 0.66 |
| Kimi-Eval | 7.1 | 1.87 |
| VoiceBench | 87.9 | 0.15 |
| AU-Harness | 3.6 (↓48.75%) | 3.65 (↑ 95.19%) |

ative to the duration of the processed audio Arriaga et al. (2024). Lower RTF is more desirable, indicating a more efficient audio evaluation framework. On the other hand, Processed Samples per Second directly quantifies the model's processing speed by measuring the average number of audio samples processed per second. It serves as a complementary measure to RTF, providing a more granular view of the model's throughput and computational efficiency.

To quantify the efficiency of existing evaluation frameworks, we conduct a study on 500 audio samples (approximately 1.05 hours) of Librispeech-test-clean (Panayotov et al., 2015). As observed in Table 2, existing audio evaluation kits exhibit high RTF and slow sample processing speed. As the number of samples continues to increase with more diverse datasets, this challenge can significantly slow down the inference progress at scale.

## 3.2 CUSTOMIZABLE EVALUATION CONFIGURATIONS

Despite the strong support for various tasks and LALMs, current LALM evaluation kits provide insufficient customizations for evaluation configurations.

**Multi-turn Dialogue Support**    Previous audio evaluation toolkits have largely been constrained to tasks centered on single-turn user interactions. However, as the field moves toward building interactive and context-aware voice assistants, the ability to evaluate multi-turn tasks becomes increasingly critical. Multi-turn evaluation enables a more realistic assessment of dialogue continuity, contextual reasoning, and the model's capacity to adapt dynamically across extended conversations. Without such support, current evaluation approaches risk overlooking key aspects of usability and robustness that are essential for next-generation LALMs in realistic agentic voice systems.

**Customizable Filtering.**    The lack of customizable filtering poses a significant barrier for researchers aiming to conduct in-depth analyses of current LALM limitations. Without the ability to refine evaluation datasets based on specific criteria, it is challenging to gain granular understanding of model performance across diverse audio conditions. For instance, while certain LALMs might perform reliably on 10-second audio chunks, they might be unable to handle short-form audio typically encountered in dialogue-state tracking systems.

**Task Hierarchical Structure & Task-Metric Aggregation.**    While DynamicSUPERB-2.0 Huang et al. (2024b) provides a comprehensive set of tasks (up to 180 tasks), it lacks mechanisms for categorizing and conducting targeted evaluation runs on specific task categories. This limitation reduces its practical value for researchers and practitioners aiming to benchmark or improve LALMs' capabilities on targeted task categories.

## 3.3 COMPREHENSIVE TASK CATEGORY COVERAGE

As demonstrated in Table 8, despite the wide coverage of tasks, existing benchmarks fail to support more complex audio reasoning and fine-grained diarization tasks.

**LLM-Adaptive Diarization.**    A key limitation of prior evaluation kits is the lack of support for diarization tasks adapted to the prompting-focused capabilities and requirements of contemporary LALMs. To address this gap, we define **LLM-adaptive Diarization** as a class of tasks aimed at identifying "who says what and when" given continuous audio inputs purely through prompting rather than neural modeling. Unlike conventional audio understanding tasks, these tasks require models to segment the audio streams and localize the timing of specific information with them. Exemplars include speaker diarization (Anguera et al., 2012) and emotion diarization (Wang et al., 2023), both of which demand precise timestamp predictions for accurate evaluation. In the context of LALMs, this poses additional challenges, particularly regarding the precision of temporal predictions — an issue frequently observed in text-based LLMs (Feng et al., 2025). As a result, LLM-Adaptive Diarization calls for the development of specialized prompting strategies and adaptive evaluation metrics tailored to the unique characteristics of LALMs beyond the traditional widely adopted Diarization Error Rate metric (Galibert, 2013).

**Spoken Language Reasoning.**    Existing benchmarks remain largely centered on the audio understanding tasks, with limited emphasis on tasks requiring deeper cognitive and reasoning abilities (Peng et al., 2024). Following the Natural Language Processing (NLP) community, we define **Spoken Language Reasoning** tasks as those that involve integrating information from multiple sources to derive new conclusions without relying solely on models' memorization, knowledge-based storage and provided context (Yu et al., 2024). Unlike existing *observational audio reasoning* benchmarks (Wang et al., 2025b; Ma et al., 2025; Kumar et al., 2025), our task suite is designed to evalu-

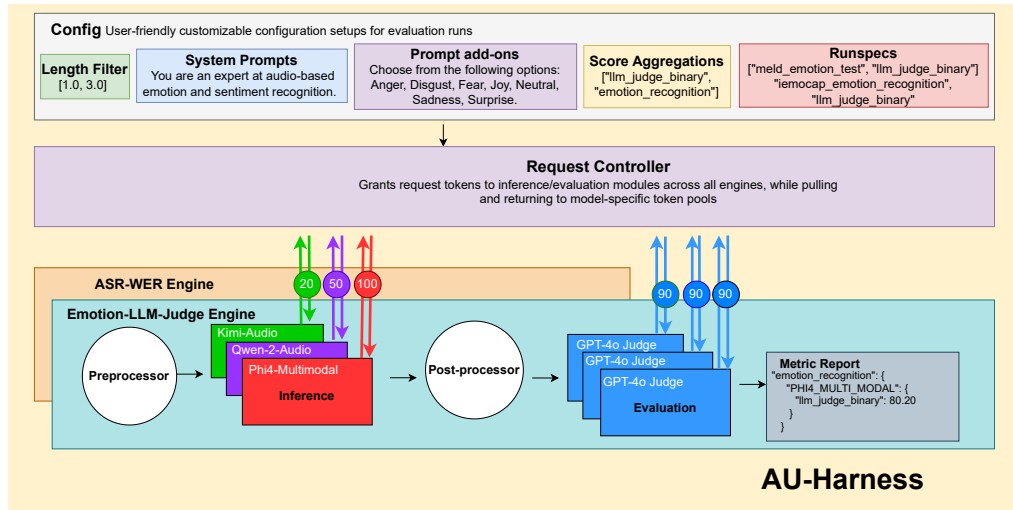

Figure 1: **Architecture overview of AU-Harness evaluation framework.** Our system comprises three core components: (1) *Config* module for hierarchical task configuration and standardized prompting, (2) *Request Controller* managing token-based concurrency limits across all engines with adaptive retry mechanisms, and (3) *Concurrent Engines* executing parallel model evaluation with dataset sharding. The Request Controller maintains a global token pool accessible to all engines, enabling efficient resource utilization and scalable throughput. Multiple concurrent connections between the controller and inference models illustrate parallel request dispatch, with each engine supporting multi-model evaluation on targeted datasets.

ate *operational reasoning* competencies, including: function calling, code generation and multi-turn complex instruction following capabilities.

## 4 AU-HARNESS

In response to the presented challenges of current audio understanding evaluation toolkits, we propose a standardized, efficient, highly customizable evaluation framework, **AU-Harness**, as detailed in Figure 1.

AU-Harness is composed of 3 primary components: **Config**, **Request Controller** and **Concurrent Engines**. The Config module defines a structured and hierarchical representation of customizable configurations, enabling flexible and transparent evaluation settings. The Request Controller is responsible for managing token requests and coordinating execution across the framework. Finally, the Concurrent Engines module carries out task-specific evaluations in parallel, where each engine can support multi-model evaluations tailored to particular tasks. In the following sections, we introduce our architecture design in detail to address the presented challenges in Section 3.

### 4.1 INFERENCE EFFICIENCY

As illustrated in Figure 1, AU-Harness maximizes inference efficiency through a token-based request scheduling architecture. More specifically, we introduce a Central Request Controller that maintains and regulates a pool of available tokens which are accessible to all models across all evaluation engines. Here, a *token* refers to a concurrency slot representing permission to issue one inference request (not a model input token), which is acquired before dispatch and released upon completion. Each concurrent engine-specific requester periodically draws from the global pool. Within each engine, multiple models are executed concurrently on a targeted dataset, with inference calls dispatched in parallel to fully exploit available computational resources. This architecture ensures that evaluation throughput is not bottlenecked by model or engine-specific constraints, but rather governed solely by user-defined request limits set globally, providing both scalability and predictable performance guarantees. Furthermore, we allow user-specified retry counts on request errors, en-

abling users to set higher request limits with the assurance that occasional failures will be re-tried and successfully completed, thereby offering a tunable balance between throughput and reliability.

Furthermore, AU-Harness implements a layered request synchronization strategy that adaptively staggers request wait times across concurrent models. This design increases the probability that all models processing a given dataset segment complete their inference in a temporally aligned manner. By reducing discrepancies in model response times, the strategy minimizes idle periods within each engine, thereby mitigating intra-engine waiting time and improving overall throughput efficiency.

Additionally, we implement dataset sharding, which partitions the evaluation dataset into disjoint subsets to enable parallel processing across multiple model endpoints. To maximize efficiency, sharding is performed proportionally to each endpoint's capacity for concurrent requests, ensuring balanced utilization of heterogeneous resources. This enables near-linear scaling of inference throughput, effectively distributing the computational workload and minimizing bottlenecks. Finally, our native integration with vLLM leverages a range of inference-level optimizations, further accelerating model execution and overall evaluation system.

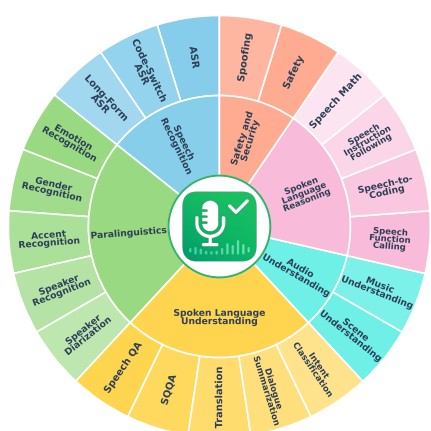

Figure 2: **Task distribution and coverage in AU-Harness.** Our framework encompasses six major categories with balanced representation. It reflects coverage from basic perception to complex reasoning, with novel emphasis on prompted temporal understanding and audio-conditioned cognitive tasks.

### 4.2 CUSTOMIZABLE EVALUATION CONFIGURATIONS

AU-Harness is highly customizable with structured task coverage as presented in Figure 2, from dataset usage to inference to evaluations.

**Decoupling Inference and Model Hosting.** AU-Harness decouples predictive inference and metric computation from model hosting infrastructure. In this way, regardless of whether the model is served through vLLM, a third-party API, or a lightweight FastAPI (Ramirez, 2018) deployment, the request handler requires only a standardized model specification to integrate seamlessly with the inference pipeline. This separation not only promotes modularity and extensibility of the evaluation framework but also enables straightforward integration via simplified future integration of alternative inference strategies.

**Wide Model Support.** AU-Harness is designed for broad model compatibility, enabling out-of-the-box evaluation across diverse inference backends. It provides native support for vLLM compatible models, which deliver high-throughput and memory-efficient inference. Models not integrated with vLLM are also supported, as long as they expose a standard /v1/chat/completions endpoint. This flexibility maximizes model coverage by enabling seamless evaluation across both vLLM- compatible and non-compatible models. To facilitate the integration, we provide boilerplate FastAPI server implementations that make it easy to build lightweight inference endpoints. Alternatively, developers can also bring their own optimized inference stacks and wrap them with FastAPI to integrate smoothly with AU-Harness, ensuring minimal overhead and maximum compatibility.

**Multi-turn Dialogue support.** By using synchronous, turn-based evaluation chains that append model outputs to the context, AU-Harness supports multi-turn evaluation of both audio and text datasets across LALMs. The simplicity and conciseness of our implementation allows for future contributions for more complex and custom multi-turn tasks as well.

**Customizable Filtering.** Great effort has been made into making AU-Harness as customizable as possible, while still being intuitive to use. First, any number of open-source and proprietary models, across any number of datasets, can be run. Each model can have its own specific temperature and max-token settings, which can override customizable, task-specific, temperature settings. Each

model endpoint also has specific run specifications that can be changed, such as concurrently allowed requests, error retry limit, timeout before retry, and audio chunk size. This maximizes resource utilization while minimizing the overall evaluation time.

**Evaluation customization.** AU-Harness is also designed for granular control over evaluation steps by allowing for customizable metric assignment on a per-dataset and per-task basis. For instance, LLM-as-judge supports configurable concurrency to maximize the throughput for evaluation stage. For a more comprehensive understanding of model performance, the framework offers configurable aggregation metrics. This capability allows for the multi-dimensional analysis of task and metric results, providing a holistic view that extends beyond simple, individual scores or sub-tasks.

### 4.3 Comprehensive Task Category Coverage

**LLM-Adaptive Diarization** Following Wang et al. (2024), we adapt diarization tasks as a special category of ASR. More specifically, to alleviate the potential issues of (1) precise temporal predictions and (2) timing mismatch of ASR and Diarization systems, we incorporate the speaker information into the transcripts and prompt LLMs to generate the ASR hypotheses. The generated hypotheses are then post-processed and evaluated on the word-level via Word-diarization Error Rate (WDER) (Shafey et al., 2019) and concatenated minimum-permutation word error rate (cpWER) (Watanabe et al., 2020). Further details of the aforementioned difference in conjunction with detailed empirical study on temporal understanding of LALMs are provided in Appendix A.5.

**Spoken Language Reasoning** Derived from text-based reasoning tasks, we introduce three novel audio-based reasoning tasks by converting the audio instructions into audio context via Text-to-Speech (TTS) system. Our work centers on 3 major reasoning tasks, including:

- **Speech Function Calling (Speech-FC)**: Speech Function Calling aims to assess the LALMs' comprehension of spoken instructions and their ability to map spoken natural language queries into structured, executable function calls with appropriate arguments. We achieve the goal by expanding BFCL-v3 (Patil et al., 2024) by systematically converting textual instructions into spoken counterparts.

- **Speech-to-Coding**: Speech-to-Coding evaluates LALMs' capability to translate spoken instructions into a formal programming language. Adapted from the renowned Spider text-to-SQL benchmark (Yu et al., 2018), we construct the Speech-Spider benchmark where LALMs are expected to convert spoken instructions into valid SQL commands.

- **Speech Instruction Following (Speech-IF)**: Proficiency in interpreting and executing intricate, potentially multi-step audio instructions is a critical skill for LALMs. To evaluate this capability, we develop Speech-MTBench benchmarks, deriving from the well-know text-based MTBench (Zheng et al., 2023).

## 5 Results & Discussion

Empirical evaluations across all task categories using our proposed AU-Harness are provided in Table 6. Following Wang et al. (2025a), we adopt GPT-4o-mini as judge for LLM-judge metrics due to its advanced capability. Further details of datasets and metrics are provided in Appendix A.1. For conciseness, we centralize the discussion on most of our introduced benchmarks shown in Table 3.

### 5.1 Inference Efficiency

**Evaluation Settings.** We perform an empirical evaluation to compare AU-Harness against existing evaluation kits: AudioBench (Wang et al., 2025a), VoiceBench (Chen et al., 2024), and Kimi-Eval (Ding et al., 2025). Our analysis focuses on the two key metrics RTF and Processed Samples per Second detailed in Section 3.1. We leverage 500 audio samples from 3 diverse datasets: librispeech-clean-test, ClothoAQA (Lipping et al., 2022), and MELD-Emotion (Poria et al., 2019) as detailed in Table 7. The evaluation is conducted on three different LALMs, including: Qwen2.5-Omni-7B Xu et al. (2025), Phi-4-Multimodal Abouelenin et al. (2025) and Voxtral-Mini-3B Liu et al. (2025). For conciseness, we report the averaged metric across all 3 LALMs. Additional runtime setups,

Table 3: **LALM performance on spoken language reasoning tasks.** We evaluate representative LALMs from different spectra: Open-source LALMs(small-sized, medium-sized, large-sized), Proprietary LALMs and Cascaded System LALMs across reasoning-focused tasks. Metrics include LLM-as-judge evaluations using GPT-4o-mini and task-specific automatic metrics. **Reasoning Avg** aggregates performance across different reasoning tasks where *Exec_Acc* of *Speech-Spider* is used for averaging calculation. Results reveal significant capability gaps, particularly in complex instruction-following scenarios. **Bold**: highest; underline: second highest. Refer to Appendix A.1.2 for metric abbreviations and detailed explanations.

| Models | | | Speech-FC BFCL_Score (↑) | | | | Speech-to-Coding EM (↑) \| Exec_Acc (↑) Speech-Spider | Speech-IF IF-Score Speech-IFEval | MTJudge (↑) Speech-MTBench | Speech Math EM (↑) Speech-GSM8K | Reasoning Avg (↑) |
|---|---|---|---|---|---|---|---|---|---|---|---|
| | simple | para | multi | multi-para | irrelevance | Avg | | | | | |
| *Small-sized Audio Language Models (<5B parameters)* | | | | | | | | | | | |
| Voxtral-Mini | 97.75 | 78.5 | 96 | 56 | 62.5 | 78.15 | 29.87 \| 61.14 | 40.02 | 63.19 | 70.05 | 62.51 |
| Qwen2.5-Omni-3B | 82.5 | 62 | 59 | 35 | 54.17 | 58.53 | 32.07 \| 58.44 | 40.91 | 58.75 | 14.10 | 46.15 |
| *Medium Sized Large Audio Language Models (5B-20B parameters)* | | | | | | | | | | | |
| Phi-4-Multimodal | 10.5 | 36.5 | 24.5 | 24.5 | 81.67 | 35.53 | 7.69 \| 39.46 | 44.51 | 65.44 | 73.54 | 51.70 |
| Qwen2.5-Omni-7B | 89.5 | 67.5 | 76 | 41 | 66.25 | 68.05 | 38.76 \| 71.73 | 50.83 | 62.88 | 84.23 | 67.54 |
| Kimi-Audio | 1.5 | 15 | 5.5 | 17 | 73.33 | 22.47 | 31.47 \| 63.84 | 46.11 | 60.88 | 72.55 | 53.17 |
| *Large Sized Large Audio Language Models (> 20B parameters)* | | | | | | | | | | | |
| Voxtral-Small | 98.25 | **87** | **97.5** | 73.5 | 77.92 | 86.83 | 40.16 \| 74.73 | 66.83 | 69.25 | 87.57 | **77.08** |
| Qwen3-Omni-30B-A3B-Thinking | 86.25 | 19 | 79 | 35.5 | 87.08 | 61.37 | **54.25** \| **79.62** | 82.38 | 75.25 | **93.56** | 78.44 |
| *Proprietary Audio Language Models* | | | | | | | | | | | |
| GPT-4o-mini-audio | 97.25 | 82 | 96 | 68 | 89.58 | 86.57 | 44.76 \| 73.13 | 70.47 | 64.06 | 87.79 | 76.40 |
| Gemini-2.5-Flash | 96.75 | **92.5** | 96 | **90.5** | 90.42 | **93.23** | 34.37 \| 77.12 | **86.28** | **75.31** | 90.52 | **84.49** |
| *Cascaded Systems* | | | | | | | | | | | |
| Whisper-Large-v3 + GPT-OSS-20B | 97.25 | 83 | 95.5 | 65.5 | 87.08 | 85.67 | 36.36 \| 74.93 | 73.72 | 68.31 | 91.58 | 78.84 |
| GPT-4o-transcribe + GPT-4.1-mini | **98.75** | 78.5 | 97 | 58 | 83.33 | 83.12 | 39.46 \| 74.13 | 66.69 | 67.06 | 90.75 | 76.35 |

namely *Sequential* and *Parallel*, to assure a comprehensive and fair comparison among all existing evaluation kits are also examined as detailed in Appendix A.2.

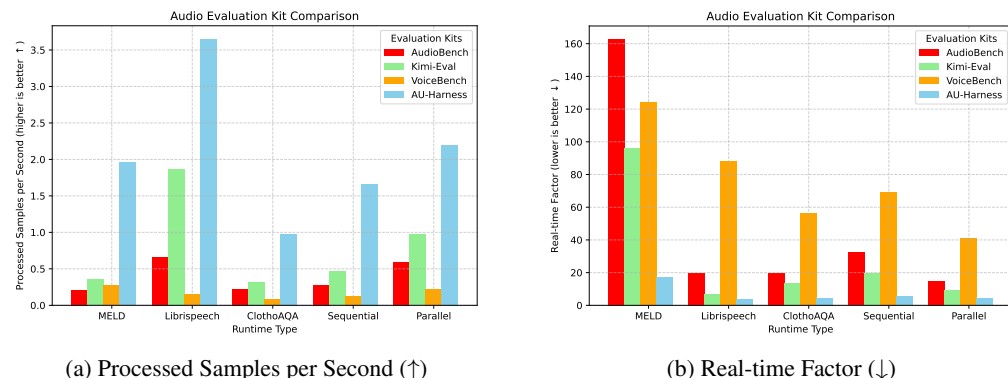

(a) Processed Samples per Second (↑)          (b) Real-time Factor (↓)

Figure 3: **Efficiency comparison across evaluation frameworks and runtime scenarios.** (a) Processed Samples per Second (↑ better) and (b) Real-time Factor (↓ better) measured across three datasets (MELD-Emotion, LibriSpeech-test-clean, ClothoAQA) and three runtime conditions: Individual (dataset-specific), Sequential (worst-case serialized execution), and Parallel (optimal concurrent execution). AU-Harness consistently outperforms existing toolkits across all scenarios, with most significant gains in parallel execution, demonstrating effective utilization of concurrent processing capabilities.

**Evaluation Comparison** As shown in Figure 3, AU-Harness consistently outperforms existing evaluation kits across all runtime scenarios in two key efficiency metrics. Specifically, AU-Harness achieves up to a 127% improvement in Processed Samples per Second and 59% reduction in RTFs compared to the next most competitive evaluation frameworks. More importantly, our *Parallel* runtime, illustrated in Figure 4, is significantly more efficient than competing frameworks. These empirical results validate our framework as a highly efficient tool for LALM evaluation.

## 5.2 INSTRUCTION MODALITY GAP

When text-based benchmarks are converted to the audio-based counterparts, the impact of instruction modality is often overlooked. However, this distinction can have a significant impact on the downstream task evaluation performance, especially for more complex instruction-following tasks. As observed in Table 4, leveraging audio instruction modality instead text can have a major impact on the performance evaluation. For instance, on challenging task of Audio Function Calling (i.e. Speech-BFCL), we observe a performance degradation of up to 7.1 points. This observation

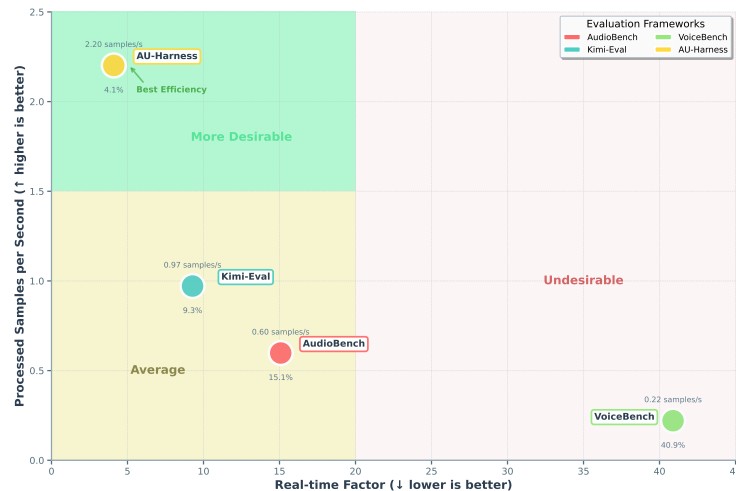

Figure 4: **Parallel runtime efficiency analysis across evaluation frameworks.** Scatter plot comparing frameworks under optimal parallel execution conditions, plotting Real-time Factor (x-axis, ↓ better) against Processed Samples per Second (y-axis, ↑ better). Our framework (rightmost cluster) achieves superior performance in both dimensions, demonstrating the effectiveness of token-based request scheduling, dataset sharding, and vLLMintegration for large-scale LALM evaluation.

Table 4: **Empirical evaluations to assess the impact of different instruction modalities on Spoken Language Reasoning tasks with Qwen3-Omni-30B-A3B-Thinking** reveals the significant performance gap between Audio and Text instructions, highlighting the need for a more thorough investigation when instruction-following benchmarks are converted from text to audio.

| Instruction Modality | Speech-IFEval (↑) | Speech-BFCL (↑) | Speech-Spider (↑) | Speech-MTBench (↑) | Speech-GSM8K (↑) | Reasoning Avg (↑) |
|---|---|---|---|---|---|---|
| **Text** | 87.56 | 68.43 | 82.12 | 81.06 | 95.3 | 82.89 |
| **Audio** | 82.38 | 61.37 | 79.62 | 75.25 | 93.56 | 78.44 |

highlights a potential core limitation of the contemporary LALMs in following audio instructions. Therefore, a careful and thorough reassessment of different instruction modality is needed to accurately measure a model's true reasoning capabilities in a multimodal context.

## 6 CONCLUSION

We introduced a modular and extensible evaluation framework for large audio-language models that emphasizes broad task coverage, ease of use, and adaptability. Its modular design enables researchers and practitioners to extend the codebase, customize benchmarks, and integrate new models or tasks without major restructuring. While efficiency gains are realized through dataset sharding proportional to endpoint capacity and seamless vLLM integration, the broader value of our framework lies in enabling flexible, large-scale evaluations that were previously difficult to conduct in a reproducible and accessible manner. By lowering the barrier to benchmarking and fostering customization, we aim to support both systematic research and practical deployment, contributing a more standardized and transparent evaluation ecosystem for LALMs.

## LIMITATIONS

**Backend dependency and reproducibility.** Our efficiency gains rely on vLLM integration, models without mature backends revert to conventional execution with reduced throughput. Support for closed-source endpoints depends on chat-completions APIs, limiting batching control and introducing provider rate limits. Even with deterministic configs, runs may vary due to endpoint queueing and transient failures, requiring documentation of capacity and request budgets for cross-institutional comparability.

**Standardization vs. task fidelity.** Standardized prompting improves reproducibility but cannot eliminate prompt sensitivity. For open-ended tasks, canonical prompts may bias results toward specific behaviors. Our LLM-Adaptive Diarization uses word-level metrics (WDER, cpWER) as proxies for temporal precision, which remains imperfect under speech overlap or rapid transitions. The community needs multiple documented prompt families and complementary temporal measures to triangulate performance fairly.

**Coverage and generalization gaps.** While we extend beyond ASR to diarization and spoken reasoning, coverage remains skewed toward English and common domains. Environmental audio, music understanding, and low-resource languages are underrepresented. Moreover, the relationship between standardized benchmark performance and real-world audio-language capabilities where contexts are noisier, more diverse, and less structured requires further empirical validation.

These limitations highlight challenges in audio-language evaluation. Achieving reproducible, comprehensive, and valid assessment requires community coordination around prompting standards, temporal diagnostics, and multilingual breadth. Our framework is designed to enable practical, systematic progress in these areas across the broader ecosystem.

## ETHICS STATEMENT

Our work focuses on responsible development of audio language model evaluation infrastructure. We have taken care to ensure that all audio datasets used in our benchmarks respect copyright and privacy guidelines, with particular attention to speaker consent in diarization tasks. While our framework enables large-scale evaluation of LALMs, we cannot guarantee that models evaluated through AU-Harness will not generate harmful or biased audio-related outputs. Researchers and practitioners are strongly encouraged to implement appropriate content filtering and bias detection when deploying LALMs in production environments. Our speech synthesis components for creating reasoning benchmarks use only publicly available, ethically sourced voice models. Additionally, we acknowledge that our current task coverage is skewed toward English and common domains, which may inadvertently reinforce existing representational biases in audio AI systems. We encourage the community to extend our framework to include more diverse languages and cultural contexts.

Regarding language model usage in manuscript preparation, we utilize them solely to refine the language used in paper to improve clarity and correctness, without generating any substantial content or claims.

## REPRODUCIBILITY STATEMENT

We are committed to full reproducibility of our evaluation framework and experimental results. All AU-Harness code, configuration files, evaluation scripts, and documentation will be publicly released under an open-source license upon acceptance. We provide comprehensive implementation details including all hyperparameters, model endpoints, dataset preprocessing steps, and evaluation metrics in our appendices. For efficiency comparisons, we document exact hardware specifications, vLLM versions, concurrent request limits, and retry policies used across all experiments. Our newly introduced reasoning benchmarks include complete details on text-to-speech synthesis parameters and prompt templates. To ensure consistent reproduction, we provide Docker containers with fixed dependency versions and detailed setup instructions for multi-node evaluation. All random seeds, sampling parameters, and LLM-as-judge configurations are specified to enable identical result replication across different research groups.

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

## A APPENDIX

### A.1 COMPREHENSIVE AUDIO EVALUATION

#### A.1.1 BENCHMARK DETAILS

We present a comprehensive benchmark suite comprising 56 diverse datasets spanning six fundamental task categories in audio and speech understanding. Our benchmark encompasses *Audio Understanding* (6 datasets), evaluating models' capabilities in audio scene analysis and music comprehension; *Paralinguistics* (12 datasets), assessing speech characteristics including emotion, gender, accent recognition, and speaker-related tasks; *Safety and Security* (2 datasets), examining robustness against adversarial inputs and spoofing; *Spoken Language Reasoning* (5 datasets), testing complex reasoning abilities from mathematical problem-solving to code generation from speech; *Spoken Language Understanding* (21 datasets), the largest category covering speech question-answering, intent classification, and translation tasks; and *Speech Recognition* (15 datasets), establishing baselines for automatic speech recognition across multiple languages and acoustic conditions.

#### A.1.2 METRIC DETAILS

- **Word Error Rate (WER)** – Measures automatic speech recognition (ASR) errors via insertions and deletions in transcribed text. Lower is better.
- **LLM-Judge (MJ)** – LLM-based evaluation of response quality. Higher is better. Reported metrics:
  - **Binary (LB)** – Binary LLM-based pass/fail correctness judgment.

Table 5: **Comprehensive Audio and Speech Datasets Overview.** Listing of 56 datasets across 6 task categories: Speech Recognition, Paralinguistics, Audio Understanding, Spoken Language Understanding, Spoken Language Reasoning, and Safety & Security.

| Task Category | Task Type | Dataset Name | Description | License |
|---|---|---|---|---|
| Speech Recognition | ASR | AISHELL-1 | High-quality Mandarin speech recognition dataset | Apache 2.0 |
| | ASR | AMI Meeting Corpus | Multispeaker meeting recordings for ASR and diarization | CC BY 4.0 |
| | ASR | CallHome (ENG) | Conversational speech corpus across multiple languages | LDC User Agreement for Non-Members |
| | ASR | Common Voice | Crowdsourced multilingual speech dataset from Mozilla | CC0 1.0 Universal |
| | ASR | FLEURS EN-US | Multilingual speech dataset for ASR and translation | CC BY 4.0 |
| | ASR | GigaSpeech | Large-scale audio and transcription corpus for end-to-end ASR | Apache 2.0 |
| | ASR | LibriSpeech | Audiobook-derived speech corpus with clean and noisy subsets | CC BY 4.0 |
| | ASR | MNSC | Large-scale multitask speech corpus | MNSC: Publicly released |
| | ASR | Multilingual LibriSpeech (MLS) | Extension of LibriSpeech with multiple European languages | CC BY 4.0 |
| | ASR | People's Speech | Large-scale open-source English speech recognition dataset | CC-BY-SA |
| | ASR | SPGISpeech | Transcriptions of financial meeting recordings | Kensho User Agreement |
| | ASR | TEDLIUM3 | Transcribed TED talks for ASR and speaker adaptation | CC BY-NC-ND 3.0 |
| | ASR | VoxPopuli | Multilingual speech corpus from European Parliament recordings | CC0 |
| | Code-switching ASR | SEAME | Mandarin-English code-switching speech dataset | LDC2015S04 |
| | Long-form ASR | Earnings21/22 | Long-form earnings call dataset for speech recognition | CC-BY-SA-4.0 |
| Paralinguistics | Accent Recognition | MNSC AR Dialogue | Dataset for accent recognition in dialogue speech | MNSC: Publicly released |
| | Accent Recognition | MNSC AR Sentence | Dataset for accent recognition in sentence-level speech | MNSC: Publicly released |
| | Accent Recognition | VoxCeleb Accent | Speech dataset with diverse speakers for accent recognition | CC BY 4.0 |
| | Emotion Recognition | IEMOCAP Emotion | Multimodal dataset for emotion recognition in speech | GPL-3.0 |
| | Emotion Recognition | MELD Emotion | Multi-party conversation dataset for emotion recognition | GPL-3.0 |
| | Emotion Recognition | MELD Sentiment | Multi-party conversation dataset for sentiment analysis | GPL-3.0 |
| | Gender Recognition | IEMOCAP Gender | Multimodal dataset for gender recognition in speech | GPL-3.0 |
| | Gender Recognition | MNSC GR Dialogue | Dataset for gender recognition in dialogue speech | MNSC: Publicly released |
| | Gender Recognition | MNSC GR Sentence | Dataset for gender recognition in sentence-level speech | MNSC: Publicly released |
| | Gender Recognition | VoxCeleb Gender | Speech dataset with diverse speakers for gender recognition | CC BY 4.0 |
| | Speaker Diarization | Callhome (ENG) | Multilingual telephone conversations for speaker diarization | CC-BY-NC-SA-4.0 |
| | Speaker Recognition | MMAU-mini | Multi-modal audio dataset for speaker recognition | Apache 2.0 |
| Audio Understanding | Music Understanding | MuChoMusic | Benchmark for music understanding for LALMs | CC-BY-SA-4.0 |
| | Scene Understanding | AudioCaps | Large-scale dataset for open-domain audio captioning | MIT |
| | Scene Understanding | AudioCaps QA | Dataset for question answering over natural audio scenes | MIT |
| | Scene Understanding | Clotho AQA | Dataset for answering natural-language questions about audio signals | MIT |
| | Scene Understanding | WavCaps | Large-scale weakly labeled dataset for audio captioning | CC-BY-NC 4.0 |
| | Scene Understanding | WavCaps QA | Large-scale dataset for audio question answering | CC-BY-NC 4.0 |
| Spoken Language Understanding | Intent Classification | SLURP | Multi-domain spoken dialogue understanding benchmark | CC-BY-NC 4.0 |
| | Speech QA | Alpaca Audio | Speech dataset for question answering with audio instructions | Apache-2.0 |
| | Speech QA | CN College Listen MCQ | Multispeaker dataset for listening-based multiple-choice questions | MERaLiON Public License |
| | Speech QA | Dream TTS MCQ | Dialogue-based multiple-choice comprehension dataset with audio | MIT |
| | Speech QA | MNSC SQA | Benchmark for reasoning and understanding in spoken language | NSC License |
| | Speech QA | OpenHermes | Speech dataset for question answering with audio instructions | CC-BY-NC |
| | Speech QA | Public-SG | Speech question answering benchmark | NSC License |
| | Speech QA | SLUE SQA | Spoken Language Understanding Evaluation benchmark | CC-BY-4.0 |
| | Speech QA | Spoken Squad | Speech dataset for extraction-based question answering | CC-BY-SA-4.0 |
| | SQQA | Big Bench Audio | Benchmark for reasoning with audio and text input | MIT |
| | SQQA | MMSU | Multi-choice question answering dataset | Apache-2.0 |
| | SQQA | OpenBookQA | Multi-choice question answering dataset | Apache-2.0 |
| | SQQA | SD-QA | Multi-choice question answering dataset | Apache-2.0 |
| | Translation | CoVoST2 (zh→en) | Large-scale multilingual dataset for speech translation | CC-BY-NC-4.0 |
| Spoken Language Reasoning | Grade School Math | GSM8k | Speech-based dataset of grade school math word problems | MIT (text dataset) |
| | Speech Function Calling | BFCL | Speech dataset for complex function calling tasks | Apache-2.0 |
| | Speech Instruction Following | IFEVAL | Speech dataset for complex instruction following | Apache-2.0 |
| | Speech Instruction Following | MTBench | Speech dataset for multi-turn instruction following | Apache-2.0 |
| | Speech-to-Coding | SPEECH_TO_SQL | Speech dataset for generating executable SQL code | Apache-2.0 |
| Safety & Security | Safety | Advbench | Speech dataset for testing resistance to adversarial or harmful prompts | Apache 2.0 |
| | Spoofing | ASVpoof2017 | Speech dataset for spoofing attack detection in real-world conditions | CC BY-NC 4.0 |

- **Detailed (LD)** – Detailed multi-level llm judgement across multiple dimensions.
  - **BigBench Audio (LBBA)** – LLM-based evaluations for BigBench-like audio tasks.
  - **RedTeaming (SafetyJudge)** – LLM-based evaluations for red-teaming and safety.
  - **MT-Bench (MTJudge)** – LLM-based evaluation for multi-turn systems.
- **BLEU** – N-gram overlap score for comparing generated and reference text. Higher is better.
- **BFCL Score** Patil et al. (2024) – Measuring structured logic form comparison between predicted and reference outputs. Higher is better.
- **SQL Score** Yu et al. (2018) – Measuring the correctness of generated SQL code in two major metrics: (1) Exact Match: generated SQL code has similar syntax as the ground truth, (2) Exec_Acc: execution accuracy of the SQL code. Higher is better.
- **Instruction Following Score (IFScore)** Zhou et al. (2023) – Measuring instruction following capability in natural language tasks via averaging accuracy across (1) strict-prompt, (2) strict-instruction, (3)loose-prompt and (4) loose-instruction scenarios.
- **Word-Diarization Error Rate (WDER)** Shafey et al. (2019) Diarization-relevant metrics whose goal is to measure the percentage of words in the transcription that has the correctly assigned speaker tag.
- **Concatenated minium-permutation Word Error Rate (cpWER)** Watanabe et al. (2020) Metric accounting for both word recognition errors and speaker attribution errors.
- **Speaker_Count_Error**: Simple calculation of error in speaker attribution given the reference and hypothesis transcripts.

Table 6: **Comprehensive LALM performance across diverse audio understanding tasks.** We evaluate three representative models: Voxtral-Mini-3B, Qwen2.5-Omni-7B, GPT-4o, and Gemini-2.5-Flash —across 19 tasks spanning Speech Recognition, Paralinguistics, Spoken Language Understanding, Audio Understanding, Spoken Language Reasoning, and Safety & Security. Metrics include standard benchmarks (WER, BLEU) and LLM-as-judge evaluations using GPT-4o-mini. Results reveal significant capability gaps, particularly in temporal reasoning (diarization) and complex instruction-following scenarios. *Performance affected by Azure OpenAI content filtering. **LB**: LLM-Judge-Binary metric, **LBBA**: LLM-Judge-Big-Bench-Audio, **LD**: LLM-Judge-Detailed, **MTJudge**: Multi-turn LLM-Judge

| Task Category | Task Name | Dataset | Metric | Voxtral-Mini-3B | Qwen2.5-Omni-7B | GPT-4o | Gemini-2.5 Flash |
|---|---|---|---|---|---|---|---|
| Speech Recognition | ASR | Librispeech | WER ($\downarrow$) | 2.1 | 1.74 | 6.25 | 2.75 |
| Paralinguistics | Emotion | MELD | LB ($\uparrow$) | 28.4 | 49.8 | 20.2 | 30.0 |
| | Gender | IEMOCAP | LB ($\uparrow$) | 54.9 | 85.8 | –* | 85.50 |
| | Accent | VoxCeleb | LB ($\uparrow$) | 13 | 28.7 | –* | 55.7 |
| | Speaker Recognition | MMAU-mini | LB ($\uparrow$) | 45.8 | 62.3 | 42 | 61.5 |
| | Speaker Diarization | CallHome | WDER ($\downarrow$) | 35.38 | 35.40 | 37.12 | 41.83 |
| Spoken Language Understanding | Spoken QA | Public-SG | LD ($\uparrow$) | 62.12 | 69.4 | 70.2 | 74.34 |
| | Spoken Query QA | Big Bench Audio | LBBA($\uparrow$) | 43.5 | 53.8 | 65 | 90.4 |
| | Speech Translation | Covost2 (zh-CN $\rightarrow$ EN) | BLEU ($\uparrow$) | 15.27 | 28.41 | 21.68 | 27.1 |
| | Spoken Dialogue Summarization | MNSC SDS(P3) | LD ($\uparrow$) | 52.2 | 52 | 61.2 | 62.8 |
| | Intent Classification | SLURP | LB ($\uparrow$) | 42.5 | 57 | 48 | 79 |
| Audio Understanding | Scene Understanding | AudioCaps QA | LD ($\uparrow$) | 14.96 | 38.4 | 15.08 | 34.82 |
| | Music Understanding | MuChoMusic | LB ($\uparrow$) | 45.4 | 59.3 | 50.2 | 72.9 |
| Spoken Language Reasoning | Speech Function Calling | Speech BFCL | BFCL_Score ($\uparrow$) | 78.15 | 68.05 | 86.57 | 93.23 |
| | Speech-to-Coding | Speech-Spider | EM ($\uparrow$) \| Acc ($\uparrow$) | 29.87 \| 61.14 | 38.76 \| 71.73 | 44.76 \| 73.13 | 34.37 \| 77.12 |
| | Speech Instruction Following | Speech MTBench | MTJudge ($\uparrow$) | 63.19 | 62.88 | 64.06 | 75.31 |
| | Speech Instruction Following | Speech IFEval | IFScore ($\uparrow$) | 40.02 | 50.83 | 70.47 | 86.28 |
| | Speech Math | Speech-GSM8K | EM ($\uparrow$) | 70.05 | 84.23 | 87.79 | 90.52 |
| Safety and Security | Safety | AdvBench | SafetyJudge ($\uparrow$) | 78.5 | 98.3 | 88.1 | 97.50 |
| | Spoofing | ASVspoof | LB ($\uparrow$) | 91.5 | 30 | 0* | 80.50 |

## A.2 INFERENCE EFFICIENCY EVALUATION SETTINGS

To provide a comprehensive and fair comparison with other evaluation kits, regardless of their underlying implementation, we introduce two additional runtime scenarios beyond individual dataset runtimes, namely *Sequential* and *Parallel*. First, *Sequential* runtime represents the most inefficient runtime by assuming each benchmark is executed in a sequential manner, where no data or model parallelization algorithms are introduced. On the other hand, *Parallel* presents the theoretical upperbound for optimal runtime. The final runtime is calculated by taking the longest runtime among all evaluated datasets. This scenario presumes an ideal, zero-overhead parallelization environment where communication protocols among parallel processes and other overheads do not impact the runtime. This is considered a best-case runtime for our framework and existing evaluation kits across all presented datasets and models.

Table 7: **Experimental setup for efficiency comparison across evaluation frameworks.** We conduct controlled experiments using 500 samples from three diverse datasets: MELD-Emotion (short emotional speech), LibriSpeech-clean (medium-length read speech), and ClothoAQA (long-form descriptive audio). Total audio duration varies from 1,476 to 11,376 seconds, enabling assessment across different audio characteristics and evaluation modalities (LLM-judge vs. traditional metrics).

|  | MELD-Emotion | Librispeech-clean | ClothoAQA |
|---|---|---|---|
| # Samples | 500 | 500 | 500 |
| Audio Duration (seconds) | 1,476 | 3,780 | 11,376 |
| Evaluation Metric | LLM-Judge | WER | LLM-Judge |

Table 8: **Comprehensive task coverage analysis across audio evaluation benchmarks.** We systematically compare task support across major frameworks spanning 2021–2025, organized by six core categories: Speech Recognition, Paralinguistics, Audio Understanding, Spoken Language Understanding, Spoken Language Reasoning, and Safety & Security. Our framework provides the most comprehensive coverage, uniquely supporting LLM-Adaptive Diarization and novel Spoken Language Reasoning tasks (Speech Function Calling, Speech-to-Coding) absent from prior work.

| Task Category | Task Name | SUPERB (2021) | DynamicSUPERB (2024) | VoiceBench (2024) | AIR-Bench (2024) | AudioBench (2025) | DynamicSUPERB-2.0 (2025) | Ours |
|---|---|---|---|---|---|---|---|---|
| **Speech Recognition** | ASR | ✓ | ✗ | ✗ | ✗ | ✓ | ✓ | ✓ |
|  | Code-switching ASR | ✗ | ✗ | ✗ | ✗ | ✓ | ✓ | ✓ |
|  | Long-form ASR | ✗ | ✗ | ✗ | ✗ | ✓ | ✓ | ✓ |
| **Paralinguistics** | Emotion Recognition | ✓ | ✓ | ✗ | ✓ | ✓ | ✓ | ✓ |
|  | Gender Recognition | ✗ | ✗ | ✗ | ✓ | ✓ | ✓ | ✓ |
|  | Accent Recognition | ✗ | ✓ | ✗ | ✗ | ✓ | ✓ | ✓ |
|  | Speaker Recognition | ✓ | ✓ | ✓ | ✓ | ✓ | ✓ | ✓ |
|  | Speaker Diarization | ✓ | ✗ | ✗ | ✗ | ✗ | ✗ | ✓ |
| **Audio Understanding** | Music Understanding | ✗ | ✗ | ✓ | ✓ | ✓ | ✗ | ✓ |
|  | Scene Understanding | ✗ | ✓ | ✗ | ✓ | ✓ | ✓ | ✓ |
| **Spoken Language Understanding** | Speech QA | ✗ | ✗ | ✓ | ✓ | ✓ | ✗ | ✓ |
|  | Spoken Query QA | ✗ | ✗ | ✓ | ✗ | ✓ | ✗ | ✓ |
|  | Speech Translation | ✗ | ✗ | ✗ | ✗ | ✗ | ✗ | ✓ |
|  | Dialogue Summarization | ✗ | ✗ | ✗ | ✗ | ✗ | ✗ | ✓ |
|  | Intent Classification | ✓ | ✓ | ✗ | ✓ | ✗ | ✓ | ✓ |
| **Spoken Language Reasoning** | Speech Function Calling | ✗ | ✗ | ✗ | ✗ | ✗ | ✗ | ✓ |
|  | Speech-to-Coding | ✗ | ✗ | ✗ | ✗ | ✗ | ✗ | ✓ |
|  | Speech Instruction Following | ✗ | ✗ | ✓ | ✗ | ✓ | ✓ | ✓ |
|  | Speech Math | ✗ | ✗ | ✗ | ✗ | ✗ | ✗ | ✓ |
| **Safety & Security** | Safety | ✗ | ✓ | ✓ | ✓ | ✗ | ✓ | ✓ |
|  | Spoofing | ✗ | ✓ | ✗ | ✗ | ✗ | ✓ | ✓ |

## A.3 CONTEMPORARY EVALUATION KITS

There are a few evaluation kits that we have built upon and been inspired by, both in evaluation framework design and task coverage.

- **AudioBench** Wang et al. (2025a): A comprehensive open-source audio evaluation framework encompassing eight core tasks and more than twenty-six curated datasets, with coverage continuing to expand. AudioBench supports both open and closed-source models and provides standardized evaluation pipelines using conventional metrics such as Word Error Rate (WER) and METEOR, alongside LLM-as-a-judge scoring for instruction-following and reasoning tasks.

- **Kimi-Eval** Ding et al. (2025): A multilingual and multi-model evaluation suite designed to assess leading Chinese and English large language models, including the Baichuan series, Qwen, GLM, and Kimi itself. The benchmark spans automatic speech recognition (ASR), multiple choice question answering (MQA), open question answering (OpenQA), and reference-based question answering (RefQA), enabling a broad assessment of both comprehension and generative audio capabilities.

- **VoiceBench** Chen et al. (2024): A focused benchmark evaluating thirty-five-plus state-of-the-art speech models across seven carefully selected datasets. While the total number of datasets is smaller than in AudioBench, the high task complexity and distinctive challenge of each dataset provide a useful test suite.

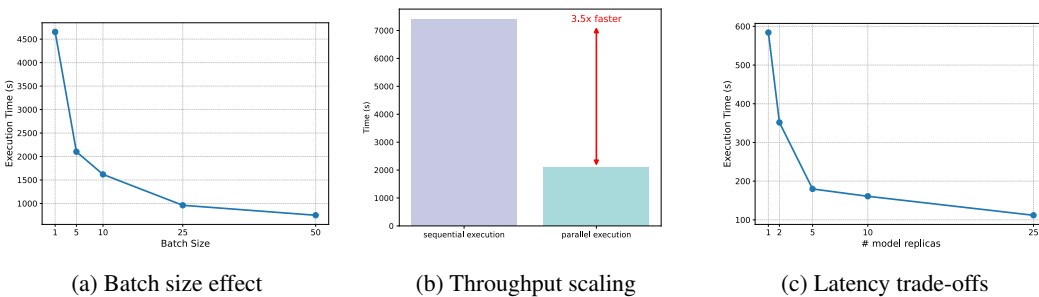

(a) Batch size effect  (b) Throughput scaling  (c) Latency trade-offs

Figure 5: **Inference efficiency ablations in AU-Harness.** We examine three factors: (a) impact of batch size on execution time, (b) throughput gains from parallel execution, and (c) latency reduction through replica scaling.

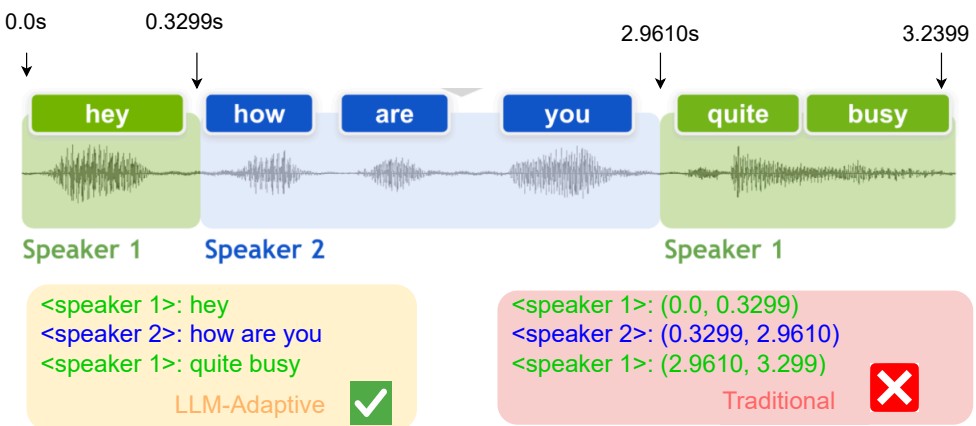

Figure 6: **LLM-Adaptive Diarization methodology comparison.** [3] Traditional diarization (top, bottom-right) outputs time-stamped audio segments with speaker annotations, ideal for specialized neural architectures. LLM-Adaptive approach (bottom-left) integrates speaker information directly into transcripts, enabling evaluation through prompting-based generation evaluated via word-level metrics (WDER, cpWER). This approach leverages LALMs' inherent language modeling capabilities while addressing temporal precision challenges through specialized evaluation protocols.

## A.4 INFERENCE EFFICIENCY ABLATIONS

To assess the scalability and efficiency of AU-Harness, we conduct three controlled ablations: (a) varying batch size, (b) throughput gains from parallel execution, and (c) latency trade-offs with replica scaling. The experimental setup follows Table 7, except for (c), where we use the full LibriSpeech-clean dataset to ensure sufficient workload for scalability analysis.

Figure 5 presents the results. Increasing batch size reduces execution time substantially, though benefits taper off at higher scales. Parallel execution yields up to a 3.5× improvement in throughput over sequential execution, confirming the efficiency of concurrent scheduling. Replica scaling further lowers latency, with near-linear improvements observed up to 25 replicas.

Overall, these ablations highlight that AU-Harness is both scalable and adaptable. By leveraging batching, parallelism, and replica scaling, it can be tuned for diverse deployment scenarios ranging from high-throughput evaluation to low-latency inference.

---

[3]Figure is adapted from NeMo documentation

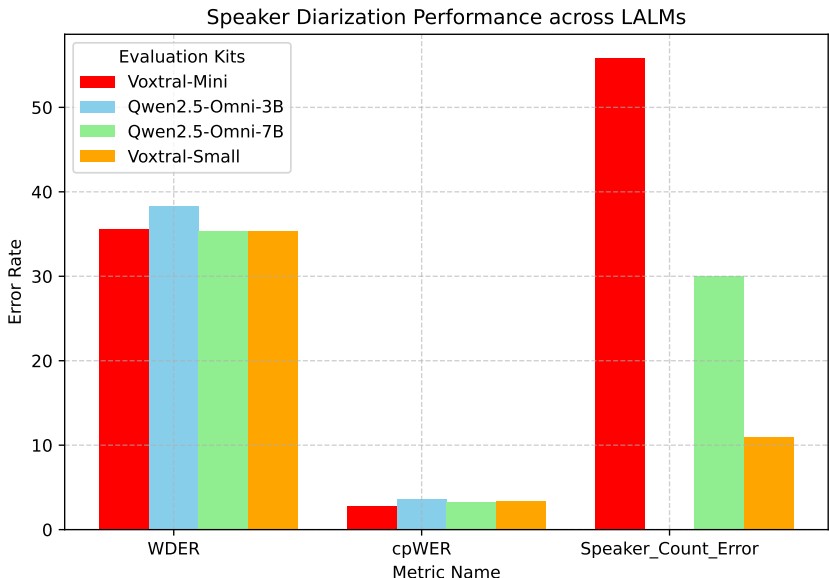

Figure 7: **LLM-Adaptive Diarization Empirical Results across LALMs.** LLM-Adaptive Diarization allows for temporal audio understanding evaluation with word-level metrics (WDER, cpWER, Speaker_Count_Error). All metrics require **"lower is better"**. Contemporary LALMs remain struggling with diarization tasks with high WDER and Speaker_Count_Error, necessitating future attention towards building and evaluating training paradigms for temporal understanding tasks.

.

### A.5  Temporal Understanding via Speaker Diarization

**LLM-Adaptive Diarization vs Neural Diarization**    Figure 6 further illustrates the key differences between *LLM-Adaptive Diarization* and *Neural Diarization* presented in Section 3.3 and 4.3. Due to the nature of LLM-prompting, precise timestamp predictions are unrealizable, especially when working with proprietary models such as Gemini-2.5 or GPT-4o.

**Empirical study of LLM-Adaptive Diarization**    As introduced in Section 4.3, we broaden the task coverage by integrating LLM-Adaptive Speaker Diarization with our proposed AU-Harness. The empirical results, as outlined in Figure 7, reveals the ongoing challenges of temporal understanding among LALMs. Specifically, *Voxtral-Mini* achieves significantly high *Speaker Count Error* metric, demonstrating its struggle with the accurate temporal localization and correct identification of speaker turn-taking in complex yet realistic audio streams. This performance gap underscores a critical area for future work, requiring future enhanced training paradigms to enhance the temporal understanding capabilities for LALMs.

### A.6  Impact of Thinking Mode of LALMs for Spoken Language Reasoning Tasks

As thinking mode might have a significant impact on reasoning tasks, we conduct further experiments to evaluate the impact of the different thinking modes on our Spoken Language Reasoning evaluation task suites. More specifically, we leverage Gemini-2.5-Flash with two different thinking modes: (1) Disabled Thinking, and (2) Dynamic Thinking. Our empirical study, observed in Table 9, reveals that Gemini-2.5-Flash achieves 7.64 absolute points of performance gain when *dynamic thinking mode* is enabled, demonstrating the essence of thinking mode on the spoken language reasoning tasks.

Table 9: **Spoken language reasoning tasks' performance on Gemini-2.5-Flash with various thinking modes** We evaluate the impact of different thinking modes of compatible LALMs on Spoken Reasoning tasks. Our empirical study reveals that enabled thinking can have a positive impact on reasoning tasks with average reasoning gain of 7.64 absolute points over no-thinking model counterpart.

| Thinking Mode | Speech-FC BFCL_Score (↑) | | | | | | Speech-to-Coding EM (↑) \| Exec_Acc (↑) Speech-Spider | Speech-IF IF-Score Speech-IFEval | Speech-IF MTJudge (↑) Speech-MTBench | Speech Math EM (↑) Speech-GSM8K | Reasoning Avg (↑) |
|---|---|---|---|---|---|---|---|---|---|---|---|
| | simple | para | multi | multi-para | irrelevance | Avg | | | | | |
| Thinking Disabled | 98 | 93 | 95.5 | 93.5 | 88.75 | 93.75 | 54.65 | 78.59 | 66.94 | 90.3 | 76.85 |
| Dynamic Thinking | 96.75 | 92.5 | 96 | 90.5 | 90.42 | 93.23 | 77.12 | 86.28 | 75.31 | 90.52 | 84.49 |

