# OpenReview forum: "AU-Harness: An Open-Source Toolkit for Holistic Evaluation of AudioLLMs"
_ICLR.cc/2026/Conference — ICLR 2026 Conference Withdrawn Submission_

### Official Review · Reviewer_Nu8r · 2025-10-30

**Soundness:** 2
**Presentation:** 2
**Contribution:** 2
**Rating:** 4
**Confidence:** 4

**Summary:**

This paper introduces an open-source framework designed to address critical challenges in the evaluation of Large Audio Language Models (LALMs). The authors identify three main limitations with existing evaluation toolkits: throughput, reproducibility, and task scope.

To overcome these issues, AU-Harness is engineered for efficiency, achieving a significant speedup over current systems. The framework also standardizes evaluation protocols by providing consistent prompting to ensure fair and reproducible comparisons between different LALMs. Furthermore, it expands the evaluative scope by incorporating a more diverse and comprehensive set of 27 audio-based reasoning tasks, aiming for a more holistic assessment of model capabilities.

The paper presents empirical results from evaluating 3 different families of LALMs, including Qwen2.5-Omni-7B, Phi-4-Multimodal, and Voxtral-Mini-3B, to demonstrate the framework's effectiveness and to highlight performance differences between those models.

**Strengths:**

1.	The motivation behind the paper is strong.
2.	Improved runtime efficiency strengthens the evaluations of LALMs.
3.	The framework structure is complete and ideally standardized, which allows other researchers to follow and build upon this work easily.

**Weaknesses:**

1.	The “380+ tasks” claim in the abstract is confusing. For instance, as mentioned in this paper, Dynamic-SUPERB Phase-2 [1] provides up to 180 tasks, where they do provide a list of them in the appendix. In contrast, this paper lists datasets and task categories but does not provide a comparable breakdown explaining how the ‘380+’ figure is derived.
2.	Limited numbers of LALMs, as only 3 open-source models are provided, while potentially omitting other viable options.
3.	See Questions.

**Reference**

[1] https://arxiv.org/abs/2411.05361

**Questions:**

1.	What experimental conditions, such as hardware specifications and software versions, were used to obtain the efficiency claims? A detailed breakdown is necessary to validate them.
2.	How were the specific tasks chosen for inclusion in the overall evaluation suite? Are these tasks collectively sufficient to ensure a holistic judgment of a LALM's capabilities?
3.	Is it sufficient to evaluate the reasoning tasks by just converting text instructions into audio context using TTS? In other words, does this experimental setup primarily test the model's transcription (ASR) capability rather than its true, deeper reasoning capacity?

---

> ### Author Response · Authors · 2025-11-26
>
> We thank reviewers for the constructive and insightful feedback on our work. We would like to address a few major questions from reviewer regarding our work:
>
> **Concerns over lack of sufficient breakdown for 380+ tasks (W1)** We appreciate reviewer's suggestions. We are going to include the detailed breakdown of 380+ tasks in our final version. For clarification, we follow the similar approach in counting the number tasks presented in DynamicSUPERB 2.0 [1]
>
> **Experimental conditions of efficient experimentation (Q1)**
> We provide detailed experimental settings in Appendix A.2 and Table 7. More specifically, we select 500 audio samples from various tasks with diverse audio durations and evaluation mechanisms. The diversity of these pre-selected tasks enables us to assess the efficiency of the evaluation engines across different task types, providing more generalizable conclusions under realistic evaluation settings.
>
> **Criteria for chosen tasks of inclusion to claim "holistic evaluation" (Q2)**
>
> _Holistic Evaluation_: For clarification, our use of the term "holistic" refers to the design of a unified and comprehensive evaluation framework. More specifically, our work aims to provide a flexible, modular, and extensible evaluation framework capable of supporting a wide range of audio-based benchmarks and models. To demonstrate the comprehensive nature of our evaluation framework, we present the supported task hierarchy in Figure 2 with the potential for future extensions. In addition, our section 4.2 "Customizable Evaluation Configurations" (L291-L333) further demonstrates the framework's holistic support by enabling fine-grained control over evaluation pipelines across diverse task settings, model deployment options.
>
> _Task Selection Criteria_: We select tasks based on the coverage level of essential task categories. Task categories include Speech Recognition, Paralinguistics, Spoken Language Understanding, Audio Understanding, Spoken Language Reasoning, Security and Safety.  These task categories highly align with the community's interest demonstrated in contemporary benchmark works [1][2][3] while supporting our newly introduced benchmarks focusing on operational reasoning.
>
> **Restrictions to 3 open-source LALMs (W2)**
> As summarized in Table 3, our evaluation covers a wide variety of current frontier LALMs. This includes: 4 open-source LALMs (Voxtral, Qwen2.5, Qwen3, Phi-4-Multimodal, Kimi-Audio) and 2 proprietary LALMs (i.e. Gemini 2.5, GPT-4o-mini-audio). Besides single LALMs, we also evaluate cascading models (with both proprietary and open-source language models). To provide further systematic coverage, we categorize the models into 5 major categories (1) Small-sized LALMs, (2) Medium-sized LALMs, (3) Large-sized LALMs, (4) Proprietary LALMs, (5) Cascaded Systems.
>
> **Reasoning evaluation:  Deep reasoning capability vs ASR capability (Q3)**
> Thank you for your insightful question. To better answer this question, we presented in our Table 4 the comparison between audio and text query form (i.e. same content in different modalities) across our operational reasoning benchmarks. We observe that there exists a performance gap between when audio or text queries are presented. This gap can be explained in two potential ways: (1) error propagation from ASR if the model tackles the challenge in a 2-stage approach (ASR + text-based LLMs), (2) challenges of complex instruction-following and reasoning when instructions are in the audio modality. Through our preliminary empirical studies, we found out that ASR performance across models on our operational reasoning benchmarks is nearly perfect. This implies the observed gap is due to the lack of deep reasoning capability of the model when instructions/queries are presented in audio modalities, not simply an error propagation from ASR in a potential two-stage system.
>
> We appreciate reviewer's feedback and suggestions. We are happy to answer any additional questions that reviewer might have.
>
> _Reference:_
>
> [1] Huang et al., 2025. Dynamic-SUPERB Phase-2: A Collaboratively Expanding Benchmark for Measuring the Capabilities of Spoken Language Models with 180 Tasks. ICLR 2025.
>
> [2] Yang et. AIR-Bench: Benchmarking Large Audio-Language Models via Generative Comprehension. ACL 2024.
>
> [3] Wang et al., 2025. AudioBench: A Universal Benchmark for Audio Large Language Models. NAACL'25

---

### Official Review · Reviewer_DfCE · 2025-10-31

**Soundness:** 1
**Presentation:** 2
**Contribution:** 1
**Rating:** 2
**Confidence:** 5

**Summary:**

This paper introduces AU-Harness, a framework for evaluating LALMs. The proposed framework provides speedup compared with existing toolkits. The paper also provides standardized prompting protocols and flexible configurations, along with two new tasks for comprehensive evaluation.

**Strengths:**

- The speedup provides better efficiency for researchers in this area.

**Weaknesses:**

- The paper lacks sufficient novelty. The main novelty is the two newly proposed tasks, while other aspects are more like engineering optimization instead of scientific innovation.
- The definition of "exisiting toolkit" is not appropriate. It seems like the so-called toolkit refers to the official implementation of the benchmark papers. However, it should be noted that the implementation of these papers should not be viewed as "toolkit" but "sample codes" instead. Therefore, for benchmark developers, it is natural that they do not focus on efficiency optimization. Therefore, the authors' statement that takes the efficiency as the weakness of exisiting works is not appropriate.
- While the paper argues that AU-Harness is holistic, the evaluation scope is quite limited. and the paper misses many important references.
- The paper only compares with three exisiting benchmarks, which is insufficient.

**Questions:**

I have listed my concerns in the weakness section. I think the paper is not novel enough, and it also misses many references to relevant prior work.

The evaluation landscape in this paper is quite limited. For example, while the paper evaluates reasoning capability, the evaluation only focuses on the content-based reasoning, which only requires the semantic understanding for reasoning. However, the acoustic-based reasoning that combines acoustic information for reasoning is also essential that distinguishes LALMs from text-based LLMs[1]. Therefore, I believe the "holistic" in the paper title is overclaimed.

[1] Yang et al., "Towards Holistic Evaluation of Large Audio-Language Models: A Comprehensive Survey", EMNLP 2025

---

> ### Author Response · Authors · 2025-11-26
>
> We appreciate reviewer's feedback for our work. We would like to clarify a few major aspects based on reviewer's questions. We are happy to answer any additional questions reviewer might have during the discussion period.
>
> **Definition and contribution as an audio evaluation toolkit (W1, W2)**
> We would like to elaborate on our definition and motivation as a toolkit. Unlike existing works, our focus is on addressing the lack of efficient processing pipelines for large-scale audio evaluations, particularly under constrained throughput conditions (L50-54). This leads to the 3 contemporary evaluation frameworks tackling these challenges:
>
> - **AudioBench [1]**: presents itself as an evaluation toolkit (Abstract and page 13 under comparison against AIR-Bench)
> - **VoiceBench [2]**: presents itself as a voice-assistant benchmark with unique tasks requiring dedicated evaluation setups.
> - **Kimi-Eval [3]**: is designed specifically as an evaluation kit to help facilitate the assessment of the audio capabilities.
>
> As we are aware that SUPERB [4], DynamicSUPERB [5], AIR-Bench [6] are proposed as a benchmark contribution, we do not include them for baseline comparison (Table 1, Table 2, Figure 4). Instead, we acknowledge their benchmark contributions via task coverage analysis (Table 8). Our further details are also provided in Appendix A.3 and Section 2 (Related Work).
>
> **Only 3 existing evaluation framework baselines (W4)**
> As previously mentioned, we are aware of three major evaluation frameworks that serve as toolkits for audio evaluation and are happy to clarify again. Kimi-Eval is restricted to supporting Kimi-Audio model family. On the other hand, AudioBench and VoiceBench lack high-throughput audio processing capabilities, resulting in substantial runtime evaluation processing at scale. We would appreciate it if Reviewer can point us to any additional audio evaluation frameworks that are relevant and comparable to our contemporary efforts.
>
> **Holistic evaluation is overclaimed. Lack of acoustic-based reasoning (W3)**
> We acknowledge that the term 'holistic' has been used for various purposes in contemporary literature. For further clarification, our use of the term "holistic" refers to the design of a unified and comprehensive evaluation framework. More specifically, our work aims to provide a flexible, modular, and extensible evaluation framework capable of supporting a wide range of audio-based benchmarks and models. To demonstrate the comprehensive nature of our evaluation framework, we present the supported task hierarchy in Figure 2 with the potential for future extensions. In addition, our section 4.2 "Customizable Evaluation Configurations" (L291-L333) further demonstrates the framework's holistic support by enabling fine-grained control over evaluation pipelines across diverse task settings, model deployment options. We would be happy to update the term accordingly to address the concern raised by reviewer.
>
> Though we acknowledge that we currently do not support acoustic-based reasoning such as MMAR [7], MMAU-PRO [8], our objective is to concentrate on the operational audio reasoning tasks that require executing instructions delivered through speech (L68). We have acknowledged the differences in our Introduction (L66-73) and Related Work (L96-109). Therefore, the contribution of additional acoustic-based reasoning is orthogonal to our major contributions to comprehensive audio evaluation frameworks and operational audio reasoning tasks. Additionally, since the introduction of MMAR, MMAU-PRO benchmarks, we have added support for these benchmarks with our evaluation framework, enabling further extensive task coverage. We seek to update our paper and release the framework publicly.
>
>
> _Reference:_
>
> [1] Wang et al., 2025. AudioBench: A Universal Benchmark for Audio Large Language Models. NAACL'25
>
> [2] Chen et al., 2024. VoiceBench: Benchmarking LLM-Based Voice Assistants. arXiv:2410.17196
>
> [3] KimiTeam et al., 2025. Kimi-Audio Technical Report. arXiv:2504.18425
>
> [4] Yang et al., 2021. SUPERB: Speech processing Universal PERformance Benchmark. Interspeech 2021.
>
> [5] Huang et al., 2025. Dynamic-SUPERB Phase-2: A Collaboratively Expanding Benchmark for Measuring the Capabilities of Spoken Language Models with 180 Tasks. ICLR 2025.
>
> [6] Yang et. AIR-Bench: Benchmarking Large Audio-Language Models via Generative Comprehension. ACL 2024.
>
> [7] Ma et al., 2025. MMAR: A Challenging Benchmark for Deep Reasoning in Speech, Audio, Music, and Their Mix. arXiv:2505.13032.
>
> [8] Kumar et al., 2025. MMAU-Pro: A Challenging and Comprehensive Benchmark for Holistic Evaluation of Audio General Intelligence. arXiv:2508.13992.

---

### Official Review · Reviewer_XViW · 2025-10-31

**Soundness:** 2
**Presentation:** 2
**Contribution:** 1
**Rating:** 0
**Confidence:** 4

**Summary:**

The paper introduces a new toolkit for benchmarking speech-aware language models (speech-in, text-out). Compared to existing toolkits, it emphasizes faster benchmarking speed, highly customizable configuration, and two additional tasks: speaker diarization and operational tasks. For the latter, the authors refer to spoken tasks that involve function calls, code generation, and instruction following.

Overall, there is no new problem formulation, no research question being addressed, and no key findings—only engineering work (though they do provide the code). Despite the engineering effort, the paper does not propose any novel technical contribution that enables faster benchmarking. To the best of my knowledge, the authors rely on existing infrastructure (e.g., vLLM) to boost the speed, rather than proposing a new parallelization tool or paradigm that would contribute to the broader field. The newly proposed tasks are not entirely novel either: speaker diarization has already been discussed in DynamicSUPERB, and instruction following in SpeechIFeval. The task coverage also does not match that of DynamicSUPERB. The only new contribution is the dataset for function calls and code generation, but this contribution is minor and would be more appropriate for a dedicated paper on the agentic capabilities of speech language models.

Despite ICLR welcoming benchmark papers, I believe this refers to works that formulate a new problem or direction, introduce novel datasets and metrics, and analyze existing models to reveal key insights, rather than purely software engineering papers. As a result, it is difficult to identify sufficient scientific contribution in this work. I suggest a reject.

**Strengths:**

- The benchmarking speed is really faster than existing benchmarks

**Weaknesses:**

- This is a software engineering work built upon existing frameworks, tools, and knowledge. It is known that applying the latest parallelization tools can improve speed; this work is simply the first to implement it. The results are good but expected.
- Two of the four newly added tasks are not novel (speaker diarization and instruction following). The former is discussed in DynamicSUPERB, and the latter in SpeechIFeval and VoiceBench.
- I do not consider software configurability a scientific contribution.
- The benchmarking speed is indeed faster, but the engineering contribution alone does not seem significant enough.

**Questions:**

No

---

> ### Author Response · Authors · 2025-11-26
>
> We thank reviewer for the feedback on our work. We would like to clarify a few major points. If the reviewer has any additional questions, we are happy to answer them accordingly.
>
> ## Concerns over software engineering efforts, lack of scientific contributions. (W1,W3,W4)
>
> **Relevance to the submission track:** We would like to highlight that our work is submitted under the "infrastructure, software libraries, hardware, systems" track which emphasizes developing the framework/ infrastructure to help accelerate machine learning research. Given the absence of unified and efficient evaluation framework for large-scale audio evaluations, our work presents an extensible evaluation framework that allows seamless integration of diverse models and benchmarks to facilitate the growing research and development of future LALMs, and allows for up-to-date comparative analysis with existing models. Therefore, we believe our contributions are highly aligned with the track we submitted under the guideline of ICLR 2026.
>
> **Model support beyond vLLM:** We support models beyond vLLM-compatible models with our presented boilerplate FastAPI server template. (L307-314). This allows for seamless integration of any LALMs independent from vLLM constraints, reinforcing the objective of AU-Harness as an extensible and unified evaluation framework.
>
> **Why the unified and efficient evaluation framework matters for audio research community:** As mentioned in L50-59, the unified evaluation framework enables equitable and reproducible comparison across models of different scales and architectures, especially with the rapid growth of frontier LALMs and audio benchmarks. In addition, most existing evaluation frameworks, including AudioBench, Kimi-Eval, suffer from inefficient processing pipelines, restricting their ability to handle the increasing volume, diversity, and computational demands of modern audio-based evaluations effectively and efficiently. Therefore, our proposed AU-Harness seeks to address this gap by providing a unified, efficient, and extensible evaluation framework for large-scale LALM assessments. We aim to continue maintaining the framework to support the release of new benchmarks and models, helping ensure comparative analysis which keeps up with new releases.
>
>
> ## Lack of novelty in introduced tasks such as Diarization (DynamicSUPERB), SpeechIFEval (VoiceBench) (W2)
> **LLM-Adaptive-Diarization:** Though diarization has been introduced in DynamicSUPERB [1] (Table 2, page 10), DynamicSUPERB lacks sufficient efforts to make diarization task and metrics compatible with generative models. For instance, DynamicSUPERB reports 8852.2% on DER metric (lower is better) on Qwen-Audio-Chat. This number is unexpected for the speaker-diarization community (the community would expect 0-100%) and lacks supporting discussion to explain the meaning behind the quantitative number. Instead, in our work, we present more in-depth analysis of the diarization tasks, more appropriate metrics for diarization as detailed in Figure 6 (Appendix A.5, Section 4.3 L336-343). We also provide detailed motivations for LLM-adaptive diarization tasks in Section 3.3 (L197-208).
>
> **Spoken Language Reasoning task:** Our contributions are in terms of Speech-FC (Speech-BFCL), Speech-to-Coding (Speech-Spider) and Speech-IF (Speech-MTBench) (L345-360) where our objectives are to maximize the faithfulness to the original text-based benchmarks while incorporating acoustic variability in audio samples. We do not claim to contribute Speech-IFEval as this benchmark was originally proposed in VoiceBench [2].
>
> _Reference:_
>
> [1] Huang et al., 2025. Dynamic-SUPERB Phase-2: A Collaboratively Expanding Benchmark for Measuring the Capabilities of Spoken Language Models with 180 Tasks. ICLR 2025.
>
> [2] Chen et al., 2024. VoiceBench: Benchmarking LLM-Based Voice Assistants. arXiv:2410.17196.

---

### Note · Authors · 2026-01-06

I have read and agree with the venue's withdrawal policy on behalf of myself and my co-authors.